# On the Hardware–Software Integration in Cryptographic Accelerators for Industrial IoT

Luigi Leonardi *, Giuseppe Lettieri †, Pericle Perazzo † and Sergio Saponara †

Department of Information Engineering, University of Pisa, Largo Lucio Lazzarino, 2, 56122 Pisa, Italy
* Correspondence: luigi.leonardi@phd.unipi.it
† These authors contributed equally to this work.

**Abstract:** Industrial Internet of Things (IIoT) applies IoT technologies on industrial automation systems with the aims of providing remote sensing, remote control, self-organization and self-maintenance. Since IIoT systems often constitute a critical infrastructure, cybersecurity risks have rapidly increased over the last years. To address cybersecurity requirements, we need to deploy cryptographic processing components which are particularly efficient, considering also that many IIoT systems have real-time constraints. Hardware acceleration can greatly improve the efficiency of cryptographic functions, but the speed-up could be jeopardized by a bad hardware–software integration, which is an aspect often underrated by the literature. Considering that modern IIoT devices often mount an operating system to fulfill their complex tasks, software influence on efficiency cannot be neglected. In this paper, we develop a software–hardware integration of various cryptographic accelerators with a Linux operating system, and we test its performance with two typical IIoT reference applications. We also discuss our design choices and the lessons learned during the development process.

**Keywords:** Industrial Internet of Things; cryptography; hardware acceleration; hardware–software integration; driver; OpenSSL





## 1. Introduction

Industrial Internet of Things (IIoT) refers to the application of IoT technologies on industrial automation systems. The aim is to provide capabilities of sensing, Internet-wide communication, intelligent processing, self-organization and self-maintenance within industrial information and control architectures. Among the numerous applications of IIoT, we can mention smart transportation, smart factories, smart grid, and so on. Real-time requirements are often necessary in IIoT, due to the need to maintain the controllability of the automation system, and due to latency criticality in general. As a result of the widespread adoption of IIoT, cybersecurity threats against these systems are rapidly increasing. The risks associated to IIoT cybersecurity incidents are particularly high, since many IIoT systems are considered critical infrastructures. To address both cybersecurity and real-time requirements, we need to deploy cryptographic processing components which are both secure and efficient. Within this context, hardware acceleration of cryptographic functionalities is paramount. However, we must also make an effort to efficiently integrate such hardware with the software chain, otherwise, the intrinsic efficiency of hardware cryptographic accelerators is wasted. This is especially important if we consider that many IIoT devices are so complex nowadays that they need an operating system running on them. The majority of papers that study efficiency aspects of hardware/software cryptographic components typically focus on software only [1–4] or hardware only [5]. Very little research is dedicated to efficient software–hardware integration.

In this paper, we study a software–hardware integration of various cryptographic accelerators (namely, the ones presented in [6–9]) with a Linux operating system. We test the performance of such an integration, and we discuss our design choices and the lessons

learned during the optimization process. As a result, we find that some implementation choices that may look reasonable when developing the hardware in isolation turn out to be sub-optimal when integrating the hardware with existing software architectures.

Our reference applications are the remote software update of industrial robots (e.g., UGVs) and the remote industrial sensing (e.g., temperature or pressure measurements in plants). We refer to such applications because they represent the extreme endpoints of the spectrum with respect to the size of messages to be secured within IIoT applications. Indeed, UGV firmware can easily reach megabytes in size, while messages sent by temperature or pressure sensors are typically less than a kilobyte.

The rest of the paper is organized as follows. Section 2 introduces the necessary preliminary concepts and compares with some related work. Section 3 describes our hardware system and our hardware–software integration. Section 4 reports and discusses our experimental results. Finally, Section 5 concludes the paper and reports the lessons learned during the optimization process.

## 2. Preliminaries and Related Work

### 2.1. Preliminaries

The Advanced Encryption Standard (AES) is a symmetric-key encryption algorithm standardized by the U.S. National Institute of Standards and Technology (NIST) in 2001 [10]. AES is a variant of the Rijndael cipher [11], which can work with different key lengths and block sizes. In standardizing AES, NIST fixed the block size to 128 bits, while the possible lengths of the key are 128, 192, and 256 bits. AES is included in the ISO/IEC 18033-3 standard, and it is currently the only cipher publicly approved by the U.S. National Security Agency (NSA) for the protection of top secret information.

SHA-2 (Secure Hash Algorithm 2) is a family of cryptographic hash functions that use the Merkle–Damgård construction with a one-way compression function that uses the Davies–Meyer structure. SHA-2 family has been standardized by NIST in 2001 [12], and it includes four hash functions with different digest sizes: SHA-224, SHA-256, SHA-384, SHA-512. SHA-224 is a truncated version of SHA-256 with a different set of initial values. The same holds for SHA-384 relatively to SHA-512.

SHA-3 (Secure Hash Algorithm 3), also called Keccak, is a family of cryptographic hash functions that use a sponge construction [13]. SHA-3 has been standardized by NIST in 2015 [14], with the aim not to supersede SHA-2, but rather to provide a backup hash function family for it. Indeed, SHA-3 provides for the same digest sizes of SHA-2 (SHA3-224, SHA3-256, SHA3-384, SHA3-512), but it uses a completely different approach than SHA-2 in such a way that it is unlikely that new attacks discovered against SHA-2 also apply to SHA-3.

From the standardization of the SHA-3 family in 2015 to now, the security industry has not shown a wide adoption of it, and the majority of applications still relies on the SHA-2 hash family. The first reason for this is that SHA-2 has not been shown to have structural weaknesses big enough to justify the switch, as instead happened in the transition from MD5 to SHA-1 and from SHA-1 to SHA-2. The second reason is that SHA-3 performs slower than SHA-2 when implemented in software. This is confirmed also by our experiments (see Section 4). However, the high level of parallelization of SHA-3 allows us to implement it on hardware accelerators (ASIC or FPGA) with better performance than SHA-2, both in terms of processing time [15] and energy consumption [16]. This is the reason why SHA-3 is expected to slowly replace SHA-2 in new developments, especially when SHA-3 hardware accelerators will be widely available on processors.

### 2.2. Openssl Library

OpenSSL [17] is a widely adopted, user-space library that provides software implementations for most of the cryptographic algorithms and protocols such as AES, RSA or TLS. The library exposes one (or more) API(s) calls for each operation; developers who

want to add cryptographic operation in their application just need to initialize the structures and invoke them in the correct order.

### 2.3. Related Work on HW Support to Security

In the current state of the art, in order to accelerate the computing of the main security functions, many hardware (HW) secure co-processors have been proposed. The proposed approaches at the state of art typically follow one of these two approaches: off-chip Trusted Platform Model (TPM) and on-chip Hardware Security Module (HSM). In the TPM approach, standardized as ISO/IEC 11889, a dedicated security chip implements in HW some security functions that are offloaded from the main application processor. The TPM secure chip is assembled in the same electronic board with the application processor chip and they communicate through a serial data interface such as I2C or SPI. TPM chips available in the market typically support secure keys storage and authentication and encryption functions. For example, the ST33GTPMAI2C chip [18] is compliant with Trusted Computing Group (TCG) TPM Library spec. 2.0 and includes: AIS-31 Class PTG2 compliant true random number generator (TRNG); FIPS compliant DRBG (Deterministic Random Bit Generator); SHA-1, SHA-2 (256 and 384 bits) and SHA-3 (256 and 384 bits) hashing; AES-128, 192 and 256 bits and Triple DES (Data Encryption Standard) 192 bits symmetric encryption; RSA key generation (1024, 2048 bits), signature and encryption; ECC (NIST P-256/384) Key generation, ECDH (Elliptic-curve Diffie–Hellman) encryption, ECDSA (Elliptic-curve Digital Signature Algorithm) signature/verification. The limitation of this approach is that the communication between the application processor and the secure chip usually represents a performance bottleneck. The second approach foresees to modify the architecture of the application processor with an on-chip peripheral that acts as a secure instruction set extension. Examples of this approach are the crypto-cell IPs from ARM. For example, the crypto-cell312 [19] is conceived to be integrated on-chip with Cortex-M or Cortex-R 32b processors in IoT applications. The crypto-cell312 achieves 200 MHz in 40 nm technology and supports in hardware HASH functions, such as SHA1, SHA2-256, HMAC (Hash-based message authentication code), Symmetric cryptography Engine for AES (Advanced Encryption Standard) with 128-bit keys and Chacha20, Public-key cryptography based on RSA and Elliptic Curve Discrete Logarithm problem. Being co-integrated on-chip with the application core the security core does not suffer from bandwidth bottlenecks as in case of off-chip TPM chips. However, the secure designer should carefully develop the driver and low-level software (SW) needed to access from the application SW the secure services provided by the HSM.

## 3. System Description

### 3.1. Hw Architecture of the Proposed System

The proposed secure system architecture for IoT applications foresees an HW architecture described in this section and a low-level SW one described in Section 3.2.

The HW architecture is shown in Figure 1. It has been designed and verified in SystemVerilog and includes the following blocks:

(i) A programmable application core based on RISC-V architecture with 64b instruction set and AXI4 interconnect vs. peripherals;

(ii) Dedicated memories (Rom and SRAM) to store secure information;

(iii) independent crypto-accelerators connected via AXI4 to sustain computing intensive secure algorithms like AES-128/256, SHA2/SHA3, RNG/DBRG and ECC-based crypto functions;

(iv) One-time-programmable (OTP) memory and physically unclonable function (PUF) for chip identification and secure boot (not shown).

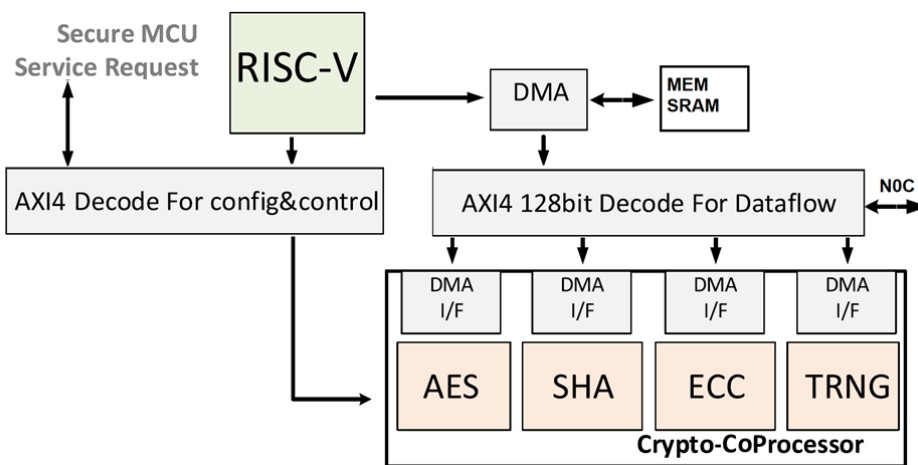

**Figure 1.** The hardware architecture of the proposed system.

The RISC-V core is a 6-stage single-issue 64b CVA6 Ariane [20] where the instruction RAM (or L1 instruction cache) has an access latency of 1 cycle on a hit, while accesses to the data RAM (or L1 data cache) have a longer latency of 3 cycles on a hit. The implementation of RISC-V used in our research includes I, M and C extensions as specified in Volume I: User-Level ISA V 2.1 as well as the draft privilege extension 1.10. A floating-point unit is also added to the ALU thus supporting from int8 to FP32 data arithmetic.

The system is completed by 4 accelerators whose detailed architectures will be discussed shortly. Each accelerator is independent from the others and has a 128-b data interface towards large data memories and a 32-b configuration interface towards the application core.

Note that, in the first generation of the system, the DMA module (shown in Figure 1) has been omitted to save circuit complexity, given the IoT target.

The AES core is detailed in [6]. It supports multiple AES-based block cipher modes, including the more advanced cipher-based MAC (CMAC), counter with CBC-MAC (CCM), Galois counter mode (GCM), and XOR-encrypt-XOR-based tweaked-codebook mode with ciphertext stealing (XTS) modes. The proposed AES accelerator implements advanced and innovative features in HW, such as AES key secure management, on-chip clock randomization to improve the resistance to side channel attacks, and access privilege mechanisms. The achieved performance when integrating the AES accelerator core in FPGA technology (Xilinx VU37P device) is the capability to encrypt or decrypt a 160B file on average in 60 μs for AES-128 and 65 μs for AES-256, i.e., for an encryption data-rate of 21.3 Mb/s for AES-128 and 19.7 Mb/s for AES-256. The AES accelerator complexity is 56 kGE (GE = gates equivalent).

The hashing core is discussed in [7]. The proposed circuit supports all the SHA2 and SHA-3 operative modes and is to be one of the hardware cryptographic accelerators within the crypto-tile of the European Processor Initiative. The accelerator has been verified on FPGA and synthesized on ASIC 7 nm TSMC silicon technology, with complexity ranging from 15 kGE (SHA2-256) to about 30 kGE (SHA2-512), and 31 kGE (SHA3-256) to about 33 kGE (SHA3-512-384-256-224). The throughput in FPGA technology is 120 ns/B to generate a 256-b hashing tag for a 8.4 kB file in SHA2. For smaller file size the throughput decreases at 210 ns/B per file of 128 B.

The other two accelerators present in the HW architecture are an RNG core described in [8], and an ECC engine described in [9]. We did not measure the performance of the HW–SW integration of such accelerators in the present paper. We leave such a task for future work.

We note that the above security throughput performance in FPGA technology refers to the specific HW accelerator, but to be really exploited from the application software, a

proper design of the accelerator driver and low-level firmware is needed. This is addressed in Section 3.2.

### 3.2. Sw Architecture of the Proposed System

The system runs the Linux kernel with the userspace based on BusyBox. The Linux Kernel, which natively supports RISC-V architecture, was not modified and its version is 5.11. It was cross-compiled, along with BusyBox version 1.33.0, using GCC version 10.2.0 and Buildroot [21] with standard optimizations. The latter is a useful tool that simplifies the configuration and the building of the two mentioned components. Due to the limited memory and processing power available on the system, the Linux Kernel was configured just enabling the required options, such as the Crypto APIs [22], Kernel Modules or the software implementation of cryptographic algorithms used in the experiments.

The kernel delegates the details of accessing each hardware device to device drivers. When a new Linux device driver for a cryptographic accelerator is developed, there are at least two approaches that can be followed: a "traditional" one, or one that follows the Linux Crypto API. The traditional method creates a special device file in the `/dev` directory; applications that want to use the device must interact with this file using the standard file systems-calls: (`read()`, `write()`, `ioctl()`, and so on). These system calls define a minimal "byte stream" abstraction, leaving all other details unspecified. Each device driver thus ends up defining its own, custom set of operations for all the device features that do not fit into the abstraction. This means that each application that wants to use any such device must include code specific for it. The burden of supporting, e.g., a new kind of accelerator device is thus left to the application developer.

The other approach, which uses the Linux Crypto API, solves this issue for cryptographic hardware devices, by adding a more expressive abstraction layer for this domain.

Figure 2 shows the entire software/hardware stack that is used in the API. When a user space application requests a cryptographic operation that is using the Crypto API, the operation is forwarded to the kernel who will dispatch it to hardware or software, according to a priority-based list. The software implementation is provided by the Linux kernel itself, as an alternative to OpenSSL. Hardware implementations usually have an higher priority over software implementations. Applications that use this API are totally unaware of what is used to compute the cryptographic operation, and thus can be accelerated, either in hardware or with a more efficient software implementation, without any change to their code. The only required action is that this new implementation must be added to the previously mentioned list. Note that the Crypto API is also used by other Linux kernel subsystems to implement features such as encrypted file systems and network communication.

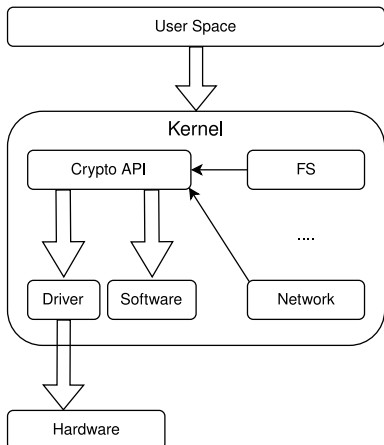

**Figure 2.** Linux Crypto API Stack.

To use the Crypto API, the driver need not create a special file in the `/dev` directory; instead, it must register itself with the kernel's list of available cryptographic algorithms.

Moreover, the driver developer must comply with the constrains posed by the Crypto APIs. The interface between the driver and the kernel defines some function callbacks that carry data and information about the operation that shall be executed. The order in which these callbacks are called, and the information they carry, are out of the control of the driver developer. This is a downside of the API, since the imposed framework may not be optimal for the hardware that may, e.g., need to perform the operations in a different order.

The Crypto API callbacks that the drier implemented must differ for each algorithm type, but the general schema is the following:

- init: it is for initializing data structures and configuring the device itself.
- update: it is the main function; here data are passed to the underlying implementation and the operation can be performed.
- exit: when the operation is over or interrupted, it can be used for reading the output (according to the crypto operation), deallocating the data structures and resetting the device.

The update function is the one that differs the most according to the algorithm type. For instance, for symmetric encryption data flow in both directions: input data are processed and then sent back to the caller. Whereas, in message digest, no output is produced at this stage.

In Listing 1 are listed the callbacks for symmetric encryption.

**Listing 1.** Crypto API Callbacks for Symmetric Encryption.

```
1  int (*setkey)(struct crypto_skcipher *tfm, const u8 *key, unsigned int keylen);
2  int (*init)(struct crypto_skcipher *tfm);
3  int (*encrypt)(struct skcipher_request *req);
4  int (*decrypt)(struct skcipher_request *req);
5  void (*exit)(struct crypto_skcipher *tfm);
```

To set up an encryption or decryption request, the AES Engine requires three information pieces:

- Key Size;
- Block cipher mode of operation: ECB, CBC, etc.;
- Encryption or Decryption.

The key size is provided to the hardware when the setkey callback is fired. The other two information pieces are given inside the init and inside the encrypt/decrypt, respectively. This has some consequences on the driver implementation because the operation can be performed only when all these three pieces are known to the hardware.

The functions are listed in the call order. The first one that is invoked by the kernel is the setkey. As a parameter of this function, the driver receives the symmetric key that will be used for encrypting or decrypting. As explained in [6], in the AES Engine, the key management system's purpose is to set and then seal all the symmetric keys at boot time in the accelerator, so that any software attack which tries to steal a key is prevented. Unfortunately, this concept cannot be applied when using the Linux Crypto APIs: each key, as far as the driver knows, is disposable and will (probably) not be reused.

Then the init function is called. Due to the way this accelerator shall be programmed, the driver cannot perform any configuration on the hardware because it does not know (yet) whether this cryptographic operation will be encryption or decryption.

The *encrypt* (or *decrypt*) function: this function is the main one. It uses two *scatterlists* for input and output data which require processing or have already been processed. As mentioned above, the device configuration must be performed inside this function because beforehand, the driver did not know what operation to perform. Once this step is complete, the driver can then proceed with all the write and subsequent read operations from the device.

As better explained in Section 4 with experiments and results, the MCU Interface used for the I/O operation for this work, which is composed of 32-bit registers for Input and Output, although fully working, is not fast enough and cannot sustain a data rate that

would allow the device to reach 100% utilization. In other words, the device spends most of its time waiting for data.

The *exit* function is called in two scenarios: when the operation is over or because the requester has suddenly canceled the ongoing cryptographic operation. The driver does not differentiate these two cases and for this reason, it resets the key slot and the entire engine, so that it is ready for a new request.

## 4. Experimental Results

All the experiments were performed on a Virtex UltraScale+ HBM VCU128 FPGA, implementing a soft-core RISC-V processor CVA6 along with the Crypto Tile cryptographic accelerator. The CPU runs at 100MHz, whilst each accelerator's engine has a different operating frequency: 170MHz AES, 190MHz SHA and 260MHz RNG. The Linux Kernel's version is 5.11 and the userspace system is based on BusyBox v1.33.0.

In order to analyze the performance of AES and SHA using OpenSSL, the Kernel's implementation and the Crypto Tile, two applications were developed: the first one using Linux's Crypto API and another one that takes advantage of OpenSSL's libcrypto implementation.

The first application was developed using libkcapi [23] v1.3.0, a library that handles all of the communications with the kernel's Crypto API interface. The application opens a file passed as an argument and maps it in memory, then saves the initial timestamp. The crypto operation is internally performed in chunks of 4096 bytes, which is the maximum allowed by the kernel. When the last operation is completed, the timestamp is saved again and the time difference, in milliseconds, is printed. For enhanced precision, the timestamp is retrieved using a special register of the RISC-V processor called *cycle*, which contains the number of clock cycles since bootstrap.

Since this application uses Linux's Crypto API, it can be used to test both the kernel's implementation and the Crypto Tile. The kernel will select the highest-priority implementation among the available ones. Since the Crypto Tile implementation has a higher priority than the software-based one, the desired selection can be forced by loading or unloading the Crypto Tile driver.

The second application was developed using the OpenSSL library v1.1.1l. The overall approach that has been followed is quite similar to the previous one: the operation is performed on chunks; this time, the size of the chunks can be chosen arbitrarily. To maximize performance, in order to compete with the accelerator, a single chunk, whose size is the entire file, was used.

In each experiment, the cryptographic operation was performed 20 times. These repetitions were not obtained by restarting the application each time. Instead, the application itself invokes the operation the required amount of times. This approach reduces all the overheads related, for instance, to the creation or destruction of a process, thus reaching a higher level of accuracy. The graphs plot the mean values thus obtained.

To understand the behavior of the algorithms and the hardware in different scenarios, the aforementioned procedure was then repeated with files of increasing size: 512 B, 5 KB, 50 KB, 500 KB and 5 MB. These sizes were chosen according to the reference applications of remote software update in robots and remote industrial sensing. Indeed, 512 B should be the typical message of an industrial sensor measuring temperature or pressure data, while 5 MB should be a credible size of a small UGV firmware. All the sizes in between should fit the messages of the majority of IIoT applications.

In Figures 3–5, the orange line represents the time taken by OpenSSL, the gray one—by the software implementation inside the kernel (Crypto/SW), and the blue one is obtained using the hardware accelerator (Crypto/HW).

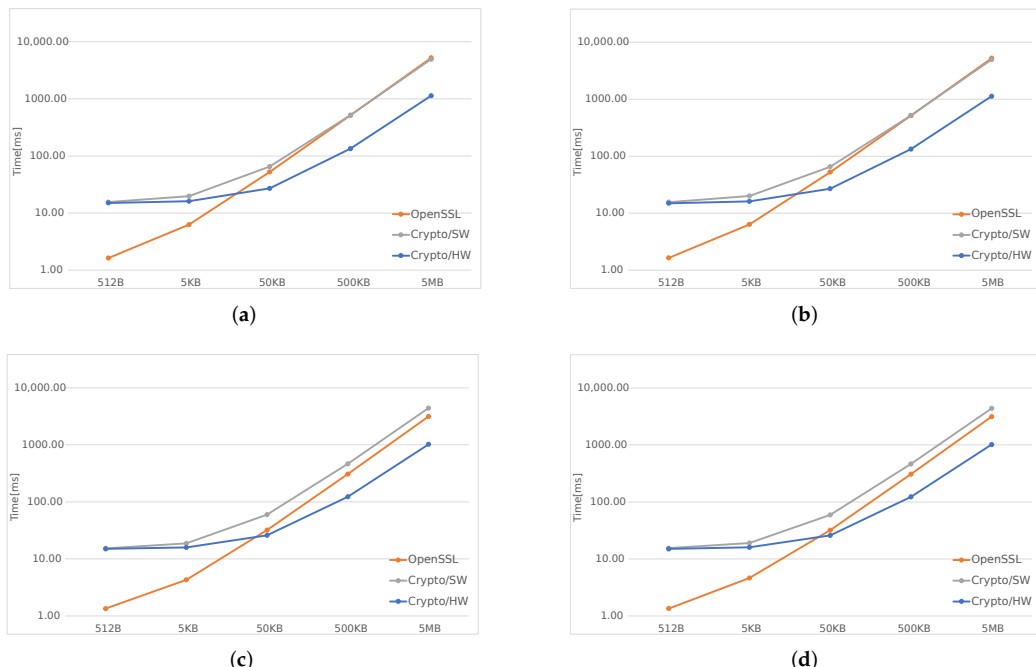

**Figure 3.** Time required to compute each SHA algorithm on increasing file sizes. (**a**) SHA-224. (**b**) SHA-256. (**c**) SHA-384. (**d**) SHA-512.

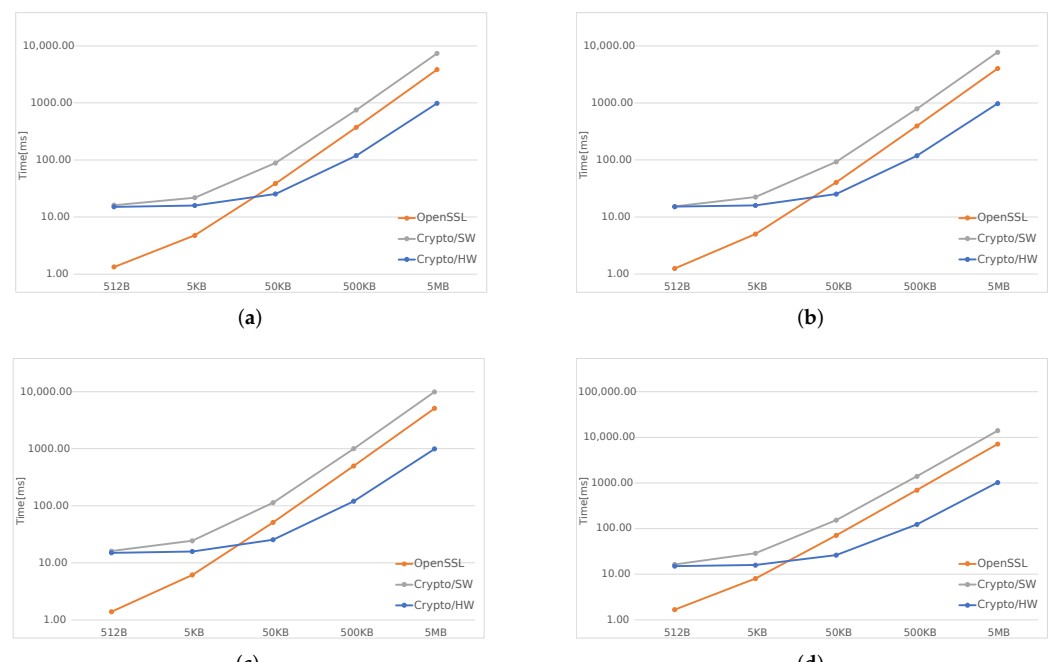

**Figure 4.** Time required to compute each SHA3 algorithm on increasing file sizes. (**a**) SHA3-224. (**b**) SHA3-256. (**c**) SHA3-384. (**d**) SHA3-512.

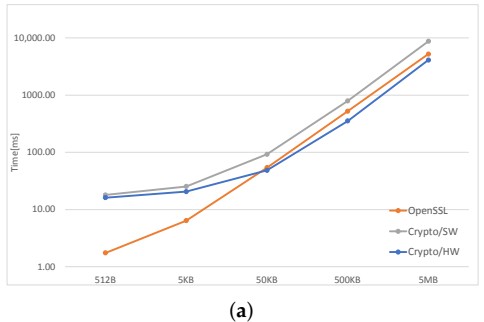
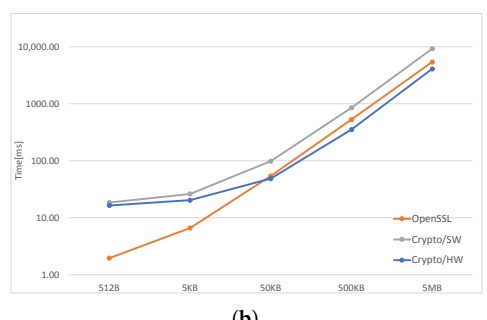

**Figure 5.** Time required to compute AES modes on increasing file sizes. (**a**) AES-ECB. (**b**) AES-CBC.

Figures 3–5, which cover all SHA algorithms and two AES modes, ECB and CBC, clearly show that, for files smaller than 50 KB both Crypto/HW and Crypyo/SW show the same performance, which is much worse than plain OpenSSL; for large files, the fastest option is Crypto/HW, while Crypto/SW remains slower than OpenSSL in most scenarios. There are several reasons behind these results: OpenSSL runs entirely in user-space, and this means that there is no Linux-related overhead when performing the computations; in particular, there is no need to cross the user/kernel interface to initialize the operation and copy the user data. This is not true for both Crypto/HW and Crypto/SW because they both live in the kernel-space, and several *syscalls* are necessary for setting up the environment, including all the data, that must be made available to the kernel, via a copy or using some other methods involving other system calls. This effect is predominant on smaller files where most of the time is spent on initialization, and this explains why the two Crypto solutions show the same performance in these cases. As the file size increases, the setup overhead is gradually amortized over larger data transfers, the gap becomes smaller and the hardware finally takes the lead, as expected. The difference between Crypto/HW and Crypto/SW for large files can be directly attributed to the hardware accelerators performance, but note that both solution still need several system calls and data copies. This may partly explain why Crypto/SW remains worse than OpenSSL even in these scenarios (this is most evident in Figure 4).

Some more experiments have been performed to more deeply understand how the hardware's time is spent. Generally speaking, the time required by an operation involving an external device can be split into at least three parts: *waiting time*, *kernel time* and *I/O time*. The first one is the time spent waiting for the operation to be completed. The second one is the time required for the data to be copied from user to kernel space, and for all the processing performed by the driver itself, excluding the I/O operations. The third one is the time spent reading or writing the data themselves.

The waiting time is always zero. The driver in all the experiments finds the output to be ready, and thus, it does not need to wait. The I/O time was not measured directly, but it was calculated: the number of iowrite and ioreads times the amount of time required for a single I/O operation. The first number can be easily obtained and depends on the file size. The only missing piece is how long it takes to perform one of these operations. This was estimated by executing a great number of I/O operations, measuring the overall time and dividing by the number of operations performed. To measure the kernel time, a modified version of the driver, named Sink, was realized removing every ioread or iowrite and then the same benchmark as before was executed. The time measured was only related to Linux and to the driver itself, who performed all the necessary calculations. Obviously, no output digest was produced.

Table 1 shows, according to the previously explained experiments, how the hardware time is spent in percentage. It is quite clear that for small files, most of the time is spent inside Linux and less doing I/O and, as the file size increases, the situation is reversed. This situation is uncommon because most of the time should be spent waiting and not inside the kernel or performing I/O.

**Table 1.** Crypto Tile time divided in sections while performing SHA3-512.

|         | 512 B   | 5 KiB   | 50 KiB  | 500 KiB | 5 MiB   |
|---------|---------|---------|---------|---------|---------|
| I/O     | 3.26%   | 23.91%  | 59.91%  | 84.36%  | 84.21%  |
| Sink    | 94.98%  | 75.29%  | 34.46%  | 14.65%  | 12.87%  |
| Waiting | 0%      | 0%      | 0%      | 0%      | 0%      |
| Total   | 98.24%  | 99.20%  | 94.37%  | 99.01%  | 97.08%  |

These results are perfectly in line regarding small file sizes, i.e., the Linux-related overhead is predominant for files smaller than 50KiB. One interesting fact is that, for bigger files, the hardware is limited by its I/O capabilities, and it could surely take less time if data were provided faster. This fact is also confirmed by Figures 6 and 7, where is plotted the time required to process a 5MB file by each SHA/AES algorithm in each of the available options. It is quite clear that the time used by the hardware is independent and constant from the chosen algorithm, while it changes when using the other two software-based alternatives, even though the computational cost of all the algorithms is different. In other words, the same pattern should also appear for the Crypto/HW section with higher/lower computation time for computationally heavier/lighter algorithms. The main cause, as explained before, is related to the I/O. The large cost of I/O also explains why the performance advantage of Crypto/HW on AES over the software solutions (Figure 5) is very small compared to SHA (Figure 3) and SHA3 (Figure 4): in AES, the bulk of the data need to travel two times from memory to the accelerator and then back, as opposed to a single time.

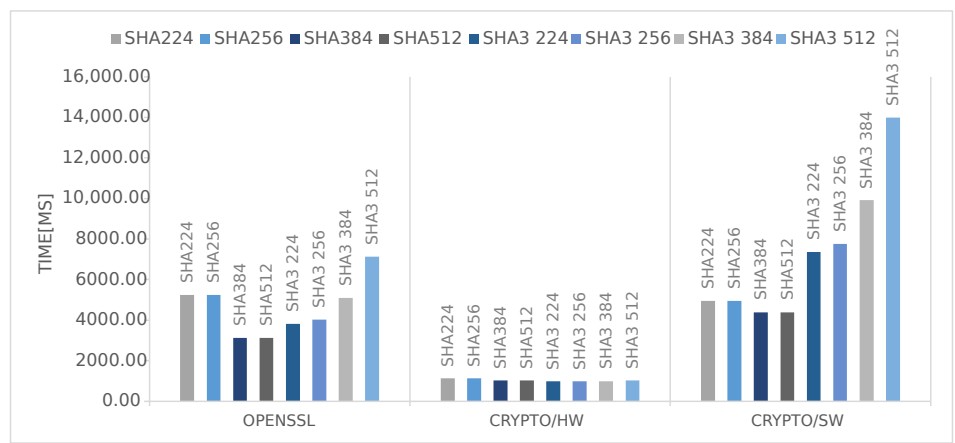

**Figure 6.** Time required by each SHA algorithm for a 5 MB file.

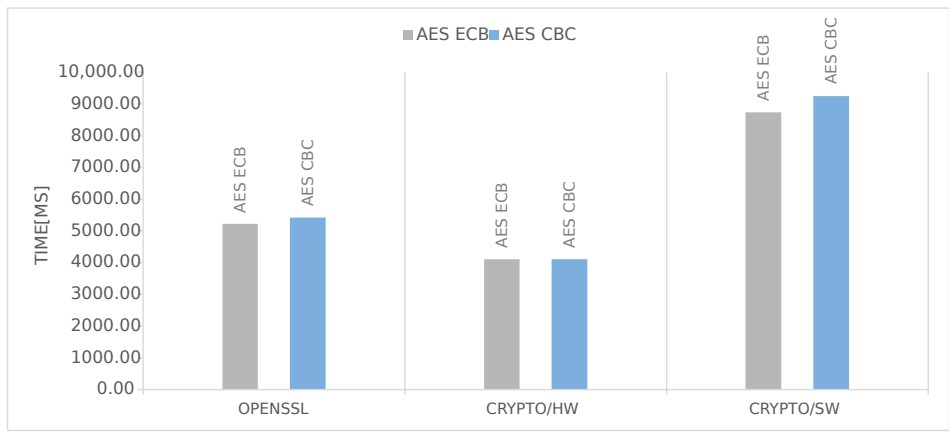

**Figure 7.** Time required by each AES algorithm for a 5 MB file.

## 5. Conclusions

We have developed a full-stack system, from hardware to Operating-System integration, for cryptographic acceleration. The implementation and the experimental results have shown that hardware–software integration must be designed carefully from the early stages of development: failure to do so may lead to unnecessary software complexity and even the impossibility to extract the full performance from the hardware. One possible approach is to follow the "hardware software co-design" approach, described in the literature [24,25]. In our case, the former problem manifested itself in the friction between the Linux Crypto API and the hardware interface, while the latter problem was caused by a sub-optimal I/O interface, which essentially limited the maximum available bandwidth. Indeed, the analysis of the experimental results has proven that, once the OS Linux kernel and drivers have been optimized, and thanks to the use of HW accelerator for crypto primitive computation, the time bottleneck in providing secure services is represented by the communication latency for data transfer. This is why in a second generation, the DMA core shown in Figure 1 has been added. The new DMA core is connected via AXI-4 memory mapped interface (with 128-bit data size so that a whole AES block can be transferred in one single transaction) to the cryptographic accelerator and to the local memory (SRAM).

Implementing a full system has also shown that the OS overhead may sometime mask the acceleration potential: since the hardware accelerators must be accessed through the inter-mediation of the OS kernel, which has a non negligible cost, cryptographic operations on small data sets may be implemented more efficiently in software. Therefore, for remote industrial sensing applications, where the typical message is very short, a solution implemented entirely in userspace software is preferable, given the cost of current kernel APIs; on the other end, for remote software updates which may involve several megabytes of data, the hardware solution can give a tenfold speed-up even when integrated in a commodity OS kernel such as Linux.

*Limitations and Future Work*

Even if the hardware/software integration were studied on a real, previously published system, our study is still limited to a single use case.

As future work, we plan to study more efficient APIs for the integration of software and crypto accelerators. For example, some of the ideas explored in network acceleration ([26]) may apply also to this domain. In addition, we plan to validate our assumption on the reference applications (e.g., the size of the messages to be secured) against the most recent literature about industrial ontology [27,28].

**Author Contributions:** Funding acquisition, S.S.; Investigation, L.L.; Software, L.L.; Supervision, G.L., P.P. and S.S.; Validation, G.L., P.P. and S.S.; Writing—original draft, L.L., G.L., P.P. and S.S.; Writing—review and editing, G.L., P.P. and S.S. All authors have read and agreed to the published version of the manuscript.

**Funding:** This work was supported by the Italian Ministry of Education, University and Research (MIUR) in the framework of the CrossLab project (Departments of Excellence), and by the European Union within the Horizon 2020 research and innovation programme "European Processor Initiative—Specific Grant Agreement 2" (grant agreement No. 101036168).

**Conflicts of Interest:** The authors declare no conflict of interest.

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
