# Peer review of "On the Hardware–Software Integration in Cryptographic Accelerators for Industrial IoT"

_applsci, doi:10.3390/app12199948_

Round 1
Reviewer 1 Report
1. The title (also part of the abstract) is not adequate for the content presented. The article is about "comparison" rather than "development" or the "integration" of cryptographic functions within the system.
2. The agenda outlined in lines 117-122 has already been presented previously in Chapter 1, and there is no point in repeating it here.
3. [Section 3.1] What is the architecture of the system? The architecture of the system must be shown in the figure. It would be interesting to know what modules and interfaces you used. For example: AXI4 interconnect consists of MemoryMapped, Lite, Stream, which is not shown. Did you use DMA, what is the communication of the hardware part with memory? Since you write about integration, the details of it must be shown.
4. [Line 139] "The RISC instruction set (...)" is an abuse. You may "The implementation of RISC-V used in our research (...)". Don't use RISC as a synonym for RISC-V, and RISC-V implementations themselves may have different extensions.
5. [Line 144] "discussed by us in previous references" It is unclear. Which "previous references"?
6. [Section 3.2] This section is actually limited to the part related to cryptography. The topic of Linux in the context of the RISC-V microprocessor is silent and this is a significant problem. Which embedded Linux was used? What about porting/compiling linux? The topic is not as well developed as for ARM microprocessors, for example, and should be better presented.
7. [Chapter 4] Graphs (a)-(j) should be signed as a common figure. Analysis of the results should refer precisely to the specific graph or graphs. The current analysis is completely illegible. There is also no reference to Figs. 2 and 3. These results are not commented on, and show an important aspect in the context of the article's goal (hardware acceleration).
8. [References] The footnotes used in lines 96, 108 and 137 should be replaced with literature references.
9. The article contains spelling errors, for example, in lines 9, 92.
Author Response
- We modified the introduction of the revised manuscript (Section 1) to clarify that the focus of the paper is on the problems arising from hardware-software integration. The comparison in Section 4 is not meant to compare different solutions, but rather to evaluate the performance of such an integration. We hope that this clarifies the current title.
- We thank the reviewer for his/her precious comment, we modified the revised manuscript accordingly.
-
The considered HW architecture includes a custom-designed cryptographic accelerator (called Crypto CoProcessor in the following figure), interconnected via memory-mapped AXI4 interface to a 64-bit RISC-V Ariane core for configuration, status monitoring and data transfer with the cryptographic accelerator. Given the IoT target, to save circuit complexity, in a first generation a DMA module has been avoided.
The OS Linux kernel, with specific drivers designed at SW side, in this work runs on the Ariane core and uses the cryptographic accelerator to speed-up computation of secure primitives like AES and SHA and others.
The achieved results prove that when optimizing the OS Linux kernel and drivers and thanks to the use of HW accelerator for crypto primitive computation, then the time bottleneck in providing secure services is represented by the communication latency for data transfer. This is why in a second generation, represented in the figure below, a DMA core has been added, which is connected via AXI-4 memory mapped interface (with 128-bit data size so that a whole AES block can be transferred in one single transaction) to the cryptographic accelerator and to the local memory (SRAM).
In the revised manuscript, we added such considerations in the conclusions (Section 6), and we added Figure 1 that illustrates the general HW architecture considered for the present paper (Section 3). We also added a sentence explaining that the DMA module is not present in the evaluated system (Section 3).
-
We thank the reviewer for his/her precious comment, we modified the manuscript accordingly.
-
Actually, the referenced line in the original manuscript was a mistake. In the revised manuscript we changed the line in “will be discussed shortly”.
-
We thank the reviewer for his/her precious comment. In the revised manuscript we added details about our compilation options used for Linux (Section 3.2). In particular we specified that we employed the BuildRoot Linux meta-distribution, compiled for ARM using gcc version 10.2.0 under standard optimizations, with the BusyBox tool bundle.
-
We modified the manuscript according to the reviewer’s suggestions (Section 4). In particular, we split the original figures into three figures and we made them larger. In addition, we further analyzed the results by clarifying why the HW and SW performance are so close for the AES algorithm. In practice this depends on the large cost of I/O operations, that in AES include two large data transfers instead of one.
-
We modified the manuscript according to the reviewer’s suggestion (Section 2).
-
We modified the manuscript according to the reviewer’s suggestion.
Reviewer 2 Report
The paper analyzed the hardware-software integration in cryptographic accelerators for industrial IoT. The work is lack of technicality. I do not see any clear research motivation of the work. Also it is not a review paper. I am afraid that I can not recommend it to be published in this journal.
Author Response
We believe that the hardware-software integration is an often underrated aspect of crypto accelerators. From our experience, without careful integration the intrinsic efficiency of hardware cryptographic accelerators is wasted. Note also that the majority of papers in the literature focus on software only or hardware only, not on the integration of the two which is an extremely important point.
In the revised manuscript we clarified the paper’s contribution (Section 1).
Reviewer 3 Report
The paper is interesting and in general well written, there are some points that need to be improved prior to publication. Please find below some comments for improving the quality and completeness of the paper:
1. The novelty of the paper is not clear. In the literature there are many papers dealing with the current topic. Authors should explicitly state in the introduction what is the novelty and the actual contribution of the paper.
2. A critical point that that authors are developing a Cryptographic methodology, but it is not mentioned for what types of data etc.
3. Since the paper deals with data in the introduction and literature review the concepts of data semantics and ontologies should be mentioned and also state how their method also applies to these technologies. As a starting point it is suggested that authors use the following reference:
a. Ameri, F., Sormaz, D., Psarommatis, F. and Kiritsis, D., 2022. Industrial ontologies for interoperability in agile and resilient manufacturing. International Journal of Production Research, 60(2), pp.420-441.
4. Some more focus should be also given to the practicality of the developed solution, why is need it, how will benefit from its usage etc, in both the introduction and conclusion.
5. Results figures do not have a caption. Also they are very difficult to read, as they have very small letters.
6. Conclusions are very poor, substantial revision is required and content enrichment. Also a small paragraph with future work and limitations is missing.
Author Response
-
In the paper we study a software-hardware integration of crypto accelerators with a Linux operating system, and we test its performance. We believe that the hardware-software integration is an often underrated aspect of crypto accelerators. From our experience, without careful integration the intrinsic efficiency of hardware cryptographic accelerators is wasted. Note also that the majority of papers in the literature focus on software only or hardware only, not on the integration of the two which is an extremely important point.
In the revised manuscript we clarified the paper’s contribution (Section 1).
-
In the paper we refer to data typically exchanged in Industrial Internet of Things applications, for example in remote reprogramming of UGV firmware and remote industrial sensing. We also refer to such applications in order to set the extremes of the message sizes tested in the experimental section. In the revised manuscript, we clearly specified the type of data we take into consideration in Section 4 (“These sizes have been chosen accordingly…”).
-
Although the ontology problem is extremely important in the Industrial Internet of Things, due to the need of formal naming of concepts and categories and the definition of relations between concepts, we believe that it is far outside the topic of the present research. In the revised manuscript, we better stated the contribution of the paper (Section 1), in order to make the main topic clearer.
-
In the revised manuscript, we better stated the reference applications within the Industrial Internet of Things scenario in Section 1 (“Our reference applications are…”).
-
We revised the manuscript according to the reviewer’s suggestions.
-
In the revised manuscript we expanded the conclusions, and we added a paragraph about limitations and future work (Section 5).
Round 2
Reviewer 3 Report
Authors have succesfully addressed the comments raised by the reviewers. The only comment that remains in my opinion is shallow analysis of the literature. It is suggested that authors find more references and depict better the domain and make more clear the reaserch gap that exists and the proposed method covers. The literature should not only focus on the actual methodology but a broader search is required for the practical use of such method
Author Response
We thank the reviewer for his/her precious comment, following your suggestion we introduced some more literature in the article, in order to provide a broader comprehension of the topics and issues introduced in the article.
For convenience, the following is the list of the introduced articles:
- Ameri, F.; Sormaz, D.; Psarommatis, F.; Kiritsis, D. Industrial ontologies for interoperability in agile and resilient manufacturing. International Journal of Production Research
- Ameri, F.; Dutta, D. An upper ontology for manufacturing service description. International Design Engineering Technical Conferences and Computers and Information in Engineering Conference
- De Michell, G.; Gupta, R. Hardware/software co-design. Proceedings of the IEEE
- Wolf, W. Hardware-software co-design of embedded systems. Proceedings of the IEEE
- Morris, J. Kernel korner: The Linux Kernel Cryptographic API. Linux Journal